# Relationship Between the Gastrointestinal Side Effects of an Anti-Hypertensive Medication and Changes in the Serum Lipid Metabolome

**DOI:** 10.3390/nu12010205

**Published:** 2020-01-13

**Authors:** Yoomin Ahn, Myung Hee Nam, Eungbin Kim

**Affiliations:** 1Department of Systems Biology, Yonsei University, Seoul 03722, Korea; dbals12345@yonsei.ac.kr; 2Environmental Risk and Welfare Research Team, Korea Basic Science (KBSI), Seoul 02855, Korea; nammh@kbsi.re.kr

**Keywords:** lipid metabolome, amlodipine, probiotics, corticosterone, ACTH, gut bacteriome

## Abstract

An earlier study using a rat model system indicated that the active ingredients contained in the anti-hypertensive medication amlodipine (AMD) appeared to induce various bowel problems, including constipation and inflammation. A probiotic blend was found to alleviate intestinal complications caused by the medicine. To gain more extensive insight into the beneficial effects of the probiotic blend, we investigated the changes in metabolite levels using a non-targeted metabolic approach with ultra-performance liquid chromatography-quadrupole/time-of-fligh (UPLC-q/TOF) mass spectrometry. Analysis of lipid metabolites revealed that rats that received AMD had a different metabolome profile compared with control rats and rats that received AMD plus the probiotic blend. In the AMD-administered group, serum levels of phosphatidylcholines, lysophosphatidylcholines, sphingomyelins, triglycerides with large numbers of double bonds, cholesterols, sterol derivatives, and cholesterol esters (all *p* < 0.05) were increased compared with those of the control group and the group that received AMD plus the probiotic blend. The AMD-administered group also exhibited significantly decreased levels of triglycerides with small numbers of double bonds (all *p* < 0.05). These results support our hypothesis that AMD-induced compositional changes in the gut microbiota are a causal factor in inflammation.

## 1. Introduction

Many metabolic diseases (e.g., obesity, hypertension, and diabetes) have emerged as serious health problems in developed countries, mainly as a result of changes in eating habits and developments in the food industry. For example, according to a report published in 2017 by the Organization for Economic Co-operation and Development (OECD), the obesity rate in the U.S. was 30.9% in 2000 and increased to 38.2% in 2014. In the case of hypertension, although prevalence decreased from 1999 to 2016, the absolute burden caused by hypertension has increased [1]. Unlike infectious diseases, metabolic disorders are typically chronic and manageable rather than remediable, forcing patients to take medications almost ad infinitum. Along with their beneficial effects, long-term medications may cause some unwanted effects. There is increasing evidence that some side effects of long-term medications, including gastrointestinal (GI) disorders (such as constipation, diarrhea, and irritable bowel syndrome (IBS)), are related to disruption of the gut microbial population, referred to as dysbiosis [2,3,4,5]. Many medications cause gut microbiota dysbiosis even though they are not considered antibiotics [6,7].

The human gut microbiota is a complex ecosystem, consisting of approximately 1–4 × 10^15^ microbial cells. The gut microbiota establishes a close relationship with the host through interactions among themselves and with host cells in the GI tract [5,8,9,10]. Hence, it seems logical rather than surprising that maintenance of a well-balanced gut microbial community is a prerequisite for healthy functioning of the whole system. Indeed, the gut microbiota is proposed to be an essential “organ” that functions to maintain nutrient metabolism, immune function, and metabolic homeostasis [11,12,13,14]. Recent studies show that the gut microbiota affects neurodevelopment and diverse brain functions by regulating the gut–brain axis, the bidirectional communication between the brain and the gut [15,16,17,18]. Many of these studies reported only correlative or associative findings; however, efforts have been undertaken to examine causality and mechanism in the microbiome.

We previously reported that amlodipine (AMD), the active ingredient in a hypertension medicine, is an aggravating factor in various bowel problems, including constipation and inflammation. This is because it induces compositional changes in the gut microbiota, since normalization of the gut microbiota alleviates intestinal complications caused by AMD [19]. To investigate the effects of the gut microbiome on the host, we performed a comparative analysis of lipid metabolome in serum samples from rats that received saline (null control), AMD, or AMD plus a probiotic blend (AMD+PB). We chose to examine lipid metabolites because they are strongly associated with high blood pressure [20], and AMD is used as a treatment for hypertension.

## 2. Materials and Methods

### 2.1. Experimental Rats

A total of 18 six-week-old male Sprague Dawley rats were randomly divided into three groups (*n* = 6/group) to receive saline (null control), AMD, or AMD+PB. The probiotic blend (PB) was obtained in powder form and consisted of *Bifidobacterium lactis* CBT BL3 (KCTC 11904BP), *Bifidobacterium longum* CBT BG7 (KCTC 12200BP), *Bifidobacterium bifidum* CBT BF3 (KCTC 12199BP), *Lactobacillus acidophilus* CBT LA1 (KCTC 11906BP), *Lactobacillus rhamnosus* CBT LR5 (KCTC 12202BP), and *Streptococcus thermophilus* CBT ST3 (KCTC 11870BP) (Cell Biotech Co., Ltd., Seoul, Korea). The PB also contained the excipients fructooligosaccharide, lactose, galactooligosaccharide, orange flavor powder, milk flavor powder, Mg-stearate, L-ascorbic acid, vitamin E, dry-formed vitamin A, vitamin B6 hydrochloride, and vitamin B1 hydrochloride. There were approximately equal numbers (ca. 1.67 × 10^9^ CFUs/g) of viable cells of each of the six bacterial strains in the PB. The total number of viable cells in the powdered form of the product was determined by measurement to be 1 × 10^10^ CFUs/g, which was diluted in water for oral administration of 1 × 10^7^ CFUs/day.

Three rats were housed in a single cage, so two cages were used for each treatment group. After a one-week acclimation period, oral gavage of PB was administered each day in a dose of ~1 × 10^7^ CFUs. Starting in the third week, AMD was administered to the rats daily for 2 weeks by oral gavage (2 mg/kg/day). The daily dose of AMD was determined as previously described [21]. All rats were housed under the following conditions: temperature 23 ± 1 °C, relative humidity 55–65%, and a 12 h light cycle. Metabolic data (weight, food intake, and water intake) were collected every day. Weight data were measured individually for each animal, but food and water intake were measured for each cage rather than for each animal. The use and care of the animals were reviewed and approved by the Institutional Animals Care and Use Committee at the Cell Biotech R&D Centre (CBTJ-15-02). All animal procedures were in accordance with the Guide for the Care and Use of Laboratory Animals issued by the Laboratory Animal Resources Commission of Cell Biotech R&D Centre.

### 2.2. Serum Collection and Serum Lipid Metabolite Analysis

On day 28, the rats were sacrificed by CO_2_ asphyxiation. It should be noted that in the AMD group one rat died before scarification. Blood samples were collected from the heart in micro tubes, kept at 4 °C for 1 h, and then centrifuged at 2200× *g* for 15 min. The supernatant was stored at −80 °C until use. Each serum sample was prepared by adding 180 μL of isopropyl alcohol to 45 μL of serum (serum:IPA, 1:4) and then vortexing for 1 min. The mixture was incubated at −20 °C for 3 h. Then, the samples were centrifuged at 14,000× rpm at 4 °C for 20 min. The supernatant was then diluted with an equal volume of deionized water and injected into an ultra-performance liquid chromatography/quadrupole time-of-flight mass spectrometry (UPLCQ/TOF–MS) machine (Waters Corporation, Milford, MA, USA). The lipid metabolites in the serum were separated using an Acquity UPLC CSH C18 column (2.1 × 100 mm, 1.7 μL particle size; Waters Corporation). The column temperature was 55 °C. The mobile phase consisted of acetonitrile:water (60:40) with 10 mM ammonium formate in 0.1% formic acid (A) and isopropanol:acetonitrile (90:10) with 10 mM ammonium formate in 0.1% formic acid (B). The flow rate was set at 0.4 mL/min. The samples were eluted using the following conditions: initial 40% B to 53% at 2 min, to 50% A at 2.1 min, to 54% B at 12 min, to 70% B at 12.1 min, to 1% B at 18 min, to 40% B at 18.1 min, followed by equilibration for an additional 2 min. Mass acquisition was performed in positive and negative electrospray ionization modes. Mass data were collected in the range of *m*/*z* 60–1400 for 20 min with a scan time of 0.25 s and an inter-scan time of 0.02 s. The source and desolvation temperatures were 120 and 550 °C, respectively.

### 2.3. Processing and Analysis of Mass Spectrometry Data

The Progenesis QI software (Waters Corporation) was used for data processing, including mass ion alignment, normalization, and peak picking. The intensities of the mass peaks for each sample were normalized according to the total ion intensity and Pareto scaled using SIMCA-P+ 12 software (Umetrics, San Jose, CA, USA).

To differentiate among the intensities of the mass peaks in each treatment group, principal component analysis (PCA) was performed. In addition, orthogonal partial least-square discriminant analysis (OPLS-DA) was used for the selection of metabolites.

Metabolites were identified by matching the measured mass spectra with references in the Human Metabolomics Database (http://www.hmdb.ca/) and METLIN (http://metlin.scripps.edu/). Lipids identified in the samples were validated on the basis of isotope similarity and fragmentation patterns. Hierarchical clustering analysis was performed using PermutMatrix (version 1.9.3, ATGC team, LIRMM, Montpellier, France) with the Pearson distance and Ward’s aggregation method.

Statistical analysis of stress hormone data and lipid metabolomic data was performed using GraphPad Prism (version 7.03; GraphPad Software, Inc., San Diego, CA, USA). Data are expressed as the mean ± SEM. The significance of differences among the data were measured by one-way ANOVA followed by Tukey’s post-hoc test, or by the Kruskal–Wallis test followed by Dunn’s post-hoc test for data that did not follow the normal distribution.

### 2.4. The Criteria for Metabolite Selection

Metabolites were selected on the basis of the following criteria: a) all differences between groups were significant (*p* < 0.05), b) the metabolite level was at least twice as high in the AMD group than in the control group and similar between the control group and the AMD+PB group, and c) the highest relative level of the metabolite was greater than 10.

## 3. Results and Discussion

### 3.1. Statistical Analysis of the Serum Metabolome

Multivariate statistical analysis of the metabolome data was performed to identify statistically significant endogenous metabolites. First, PCA was conducted to determine the inherent similarities in the spectral profiles of the treatment groups. As shown in Figure 1, the control group and the AMD group were clearly divided into two clusters on the PCA score plot, whereas the AMD+PB group displayed a pattern almost identical to that of the control group. This result is in good agreement with the previous finding that the PB alleviated intestinal complications caused by AMD [19].

### 3.2. Screening and Identification of Candidate Markers for Lipid Metabolites

To initially distinguish the differences among serum metabolites, hierarchical clustering analysis was performed to identify metabolites that were significantly increased or decreased among the treatment groups (Figure 2). Table 1 summarizes a detailed subgroup analysis of the metabolites. The G1a subgroup included cholesterol esters (CEs [22:6] and [20:4]), sterol derivatives, sphingomyelins (SMs), lysophosphatidylcholines (LysoPCs [18:0] and [16:0]), several phosphatidylcholines (PCs), and cholesterol. The G1b subgroup included PCs (18:0/22:6) and several triglycerides (TGs) with more than 10 double bonds (e.g., TG [60:12]). G1 metabolite levels overall were strongly increased in the AMD group but recovered in the AMD+PB group to the same level as those in the control group. On the other hand, the G2 metabolites included several TGs and diglycerides (DGs) with less than five double bonds (e.g., DG [34:1]). In addition, the TGs in G2 included monosaturated species (TG [48:1], TG [52:1], and TG [50:1]).

PCs and LysoPCs regulate immune function. PCs inhibit the TNF-α-induced upregulation of pro-inflammatory cytokines [22,23] and stimulate universal anti-inflammatory effects in the liver [24]. In contrast to the PCs, research on the immunomodulatory functions of LysoPCs shows conflicting results. Some studies show that LysoPCs contribute to the progression of inflammation by upregulating IL-1β-induced inducible nitric oxide synthase (NOS) [25] and also act as a death effector in the lipo-apoptosis of hepatocytes [26], which are key cells in innate immunity [27]. In addition, LysoPCs are involved in cardiovascular complications related to diabetes, rheumatoid arthritis, and atherosclerosis [28,29], as well as the activation of inflammatory responses via the acceleration of endothelial chemokine secretion [29,30]. However, other studies suggest that LysoPCs regulate inflammatory responses by inhibiting the secretion of pro-inflammatory cytokines such as TNF-α [31]. LysoPCs were evaluated as a biomarker because PC is converted to LysoPC by phospholipase A2 under inflammatory conditions [32,33,34]. Some studies suggest that LysoPCs are immunoregulatory lipid messengers under normal and pathogen-induced physiological conditions [35] because they can mediate signaling through G-protein-coupled receptors and be recognized as autoantigens [36]. Notwithstanding ambiguous results concerning pathways and mechanisms, it is certain that LysoPCs are involved in inflammation. Accordingly, it is apparent that increases of phospholipids such as PC and LysoPC in AMD-administered rats are associated with inflammation.

TGs are associated with the immune system. An excess of TGs causes diseases like hypertriglyceridemia [37], which is related to systemic inflammation [38]. Sterols play an essential role in countless biological processes including reproduction, metabolism, development, and immunity [39]. Cholesterols contribute to protection against infection by amplifying the inflammatory response and are the precursors of steroid hormones (including sex hormones, growth hormones, and glucocorticoids like corticosterone) [40]. However, excessive or prolonged cholesterol-induced immune responses can cause chronic inflammatory diseases like atherosclerosis [41]. Therefore, we performed additional analysis to examine and compare the levels of two hormones, adrenocorticotropic hormone (ACTH) and corticosterone, which are both representative stress hormones associated with immune reaction in rats [42,43,44].

### 3.3. Identification and Comparison of Corticosterone and Adrenocorticotropic Hormone (ACTH)

The probiotic blend used in this study was previously shown to have beneficial effects on human subjects with irritable bowel syndrome [45] and on animals with indomethacin-induced small intestine injury [46]. In our previous experiment, it was found to bring down increased levels of inflammatory cytokines in AMD-administered rats [19]. Glucocorticoids, including corticosterone in rodents and cortisol in humans, are anti-inflammatory steroid hormones [47,48]. In this context, we hypothesized that the probiotic blend could normalize potential anomalies in the level of corticosterone.

As shown in Figure 3, corticosterone levels were much higher in the AMD group than in the other two groups. The stress related to handling by the investigators was almost the same among the groups. Like the AMD group and the AMD+PB group, the control group was also subjected to oral gavage. Because the stress from oral gavage was the same among the groups, it is reasonable to hypothesize that higher corticosterone levels in the AMD group were caused by AMD-induced activation of the hypothalamic-pituitary-adrenocortical (HPA) axis or by a direct effect of AMD on the adrenal cortex, either or both of which were blocked by the PB co-treatment. In contrast, the ACTH levels decreased slightly more in the AMD group than in the control group (Figure 4). Considering that ACTH has a short half-life in plasma [49] and corticosterone itself is a negative regulator of ACTH secretion, the observed reduction of ACTH is likely a reflection of feedback inhibition of the HPA axis by corticosterone [47].

## 4. Conclusions

Composition and stability of the gut microbiome is known to be affected by nutrition and disease, as well as antibiotics or medication [50]. Gut microbes influence lipid processing of hosts by engaging in gene expression related to the host’s cholesterol and TG metabolism [51]. In this study, we revealed the change of lipid profiles in the serum of AMD-administered rats. Considering that impairment of the fine balance between gut microbes and the host’s immune system leads to systemic inflammation [52], it can be postulated that the change of serum lipid profiles by AMD may reflect the disturbance of the gut microbial environment by AMD. Combined with these facts, our results suggest that AMD-induced dysbiosis leads to inflammation and changes in metabolic pathways, which in turn promotes the secretion of corticosterone to relieve the symptoms (Figure 5).

## Figures and Tables

**Figure 1 nutrients-12-00205-f001:**
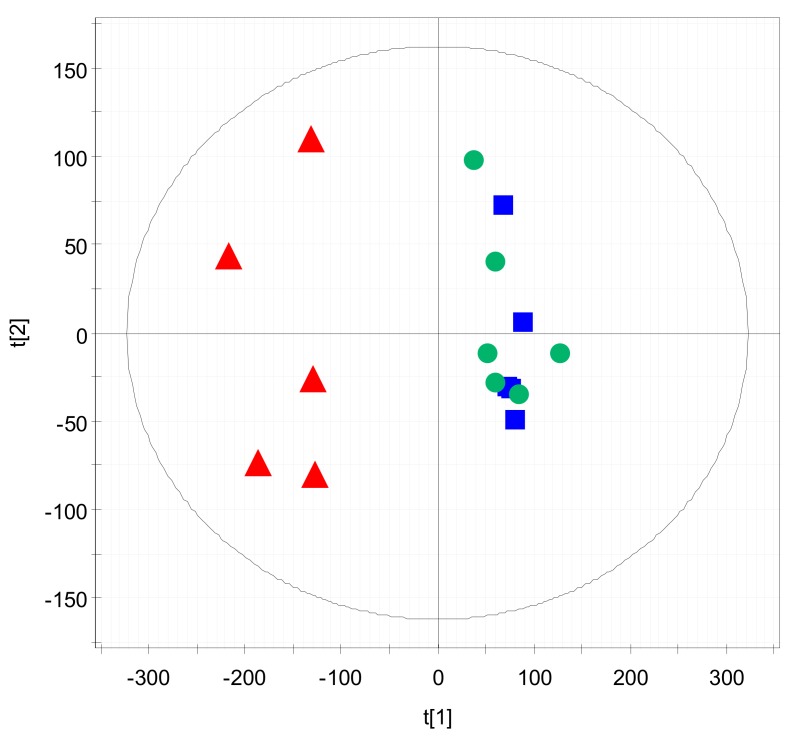
Principal component analysis (PCA) score plot of the metabolome analysis of the three treatment groups. Red triangles (▲): anti-hypertensive medication amlodipine (AMD) group. Blue squares (■): AMD plus a probiotic blend (AMD+PB) group. Green circles (●): control group.

**Figure 2 nutrients-12-00205-f002:**
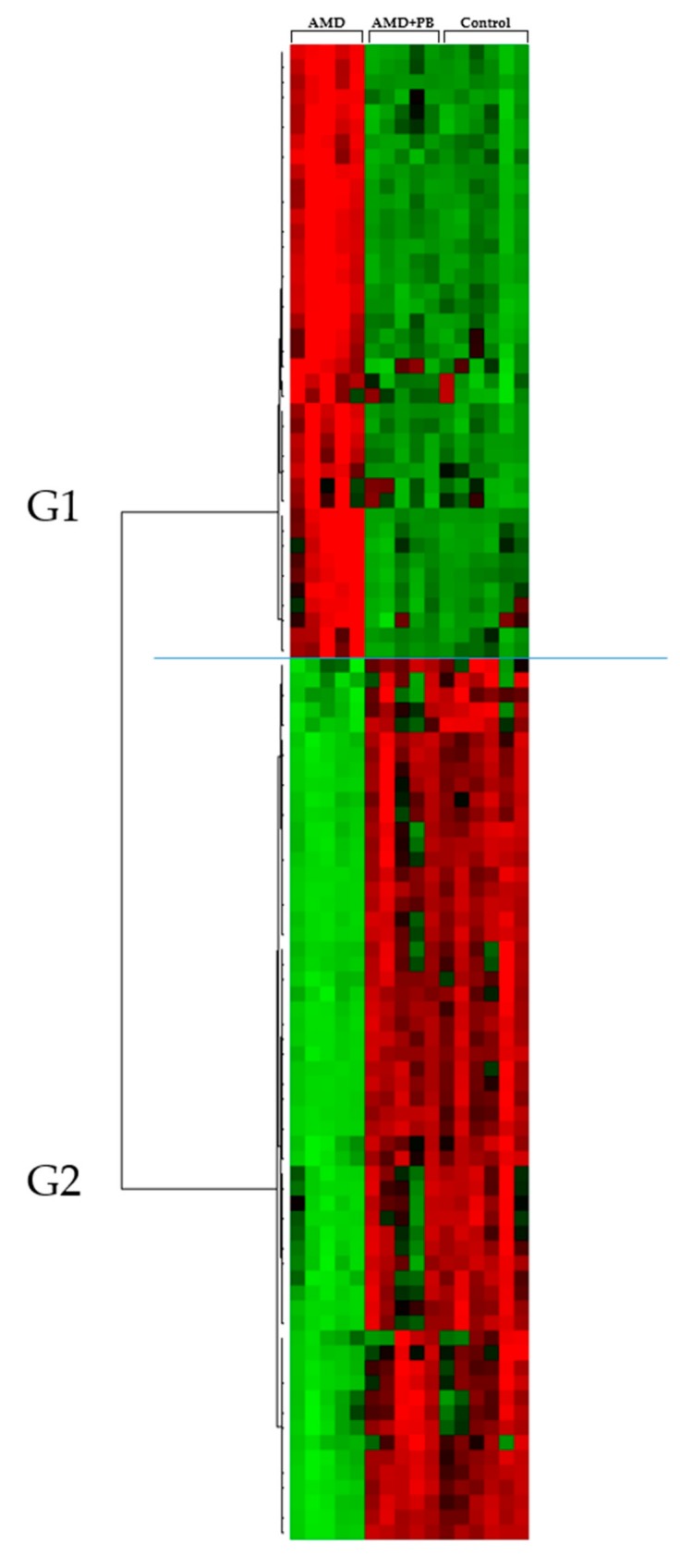
Hierarchical clustering analysis of the UPLC-HDMS metabolomics results. The rows display the metabolites, and the columns display the samples. Metabolites that significantly decreased relative to the average level across the samples are displayed in green, while those that significantly increased are displayed in red. The brightness of each color corresponds to the intensity of the difference compared with the average value.

**Figure 3 nutrients-12-00205-f003:**
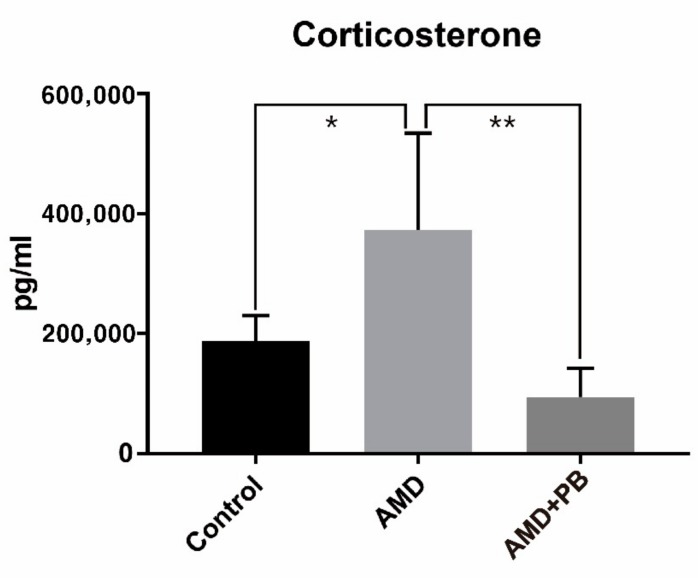
Corticosterone levels in the rat sera (*; *p* value < 0.05, **; *p* value < 0.01).

**Figure 4 nutrients-12-00205-f004:**
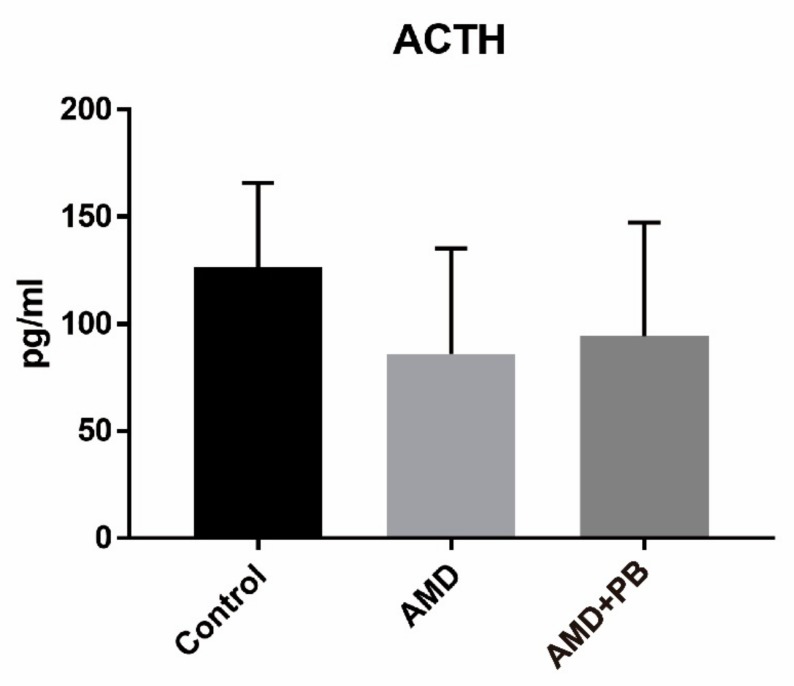
Adrenocorticotropic hormone (ACTH) levels in the rat sera.

**Figure 5 nutrients-12-00205-f005:**
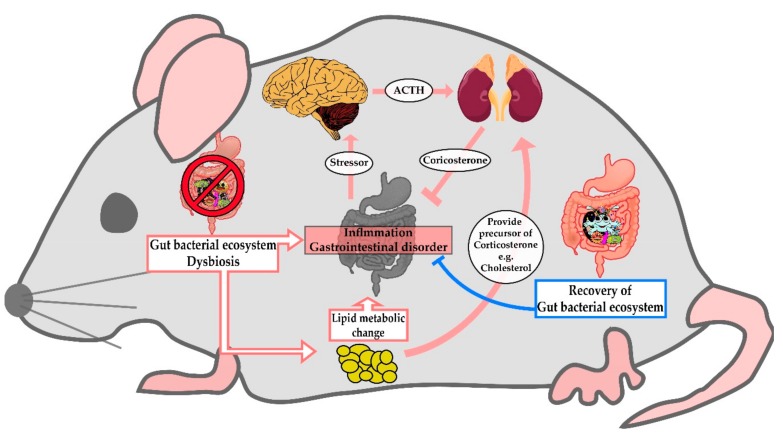
A cartoon summarizing the hypothesized effects of AMD-induced dysbiosis on lipid metabolism.

**Table 1 nutrients-12-00205-t001:** List of identified metabolites.

Group	Identification	Mean			*p* Value		Fold Change	
		C	A	A+P	A/C	A/A+P	A/C	A/A+P
G1	CE (22:6)	6 ± 3	57 ± 20	7 ± 1	0.004	0.005	8.97	8.38
	CE (20:4)	12 ± 3	43 ± 7	14 ± 2	0.000	0.001	3.69	3.08
	Sterol derivatives	42 ± 13	135 ± 37	49 ± 8	0.003	0.005	3.22	2.77
	SM (d18:1/24:1)	120 ± 27	343 ± 66	122 ± 17	0.001	0.001	2.86	2.82
	SM (d16:1/18:0)	207 ± 44	563 ± 110	223 ± 44	0.001	0.001	2.72	2.52
	LysoPC (18:0)	894 ± 160	2136 ± 356	840 ± 50	0.001	0.001	2.39	2.54
	PC (18:0/20:4)	1561 ± 302	3524 ± 573	1830 ± 339	0.001	0.001	2.26	1.93
	PC (16:0/18:0)	66 ± 11	148 ± 27	73 ± 12	0.001	0.002	2.24	2.04
	PC (16:0/16:0)	54 ± 8	118 ± 14	62 ± 10	0.000	0.000	2.17	1.91
	Cholesterol	217 ± 53	445 ± 112	229 ± 17	0.007	0.012	2.05	1.94
	LysoPC (16:0)	1484 ± 205	2893 ± 284	1385 ± 131	0.000	0.000	1.95	2.09
	TG (60:12)	25 ± 9	201 ± 64	22 ± 13	0.003	0.003	8.20	9.32
	TG (60:11)	56 ± 15	221 ± 75	43 ± 20	0.007	0.005	3.92	5.20
	TG (58:10)	88 ± 26	287 ± 71	83 ± 25	0.002	0.002	3.28	3.46
	TG (60:10)	39 ± 6	108 ± 33	30 ± 10	0.009	0.005	2.74	3.64
	PC (18:0/22:6)	279 ± 75	687 ± 244	285 ± 23	0.018	0.021	2.46	2.41
G2	TG (48:2)	159 ± 18	16 ± 6	175 ± 51	0.000	0.002	0.10	0.09
	TG (48:1)	112 ± 16	15 ± 8	124 ± 51	0.000	0.008	0.14	0.12
	TG (51:2)	148 ± 15	21 ± 7	155 ± 20	0.000	0.000	0.14	0.14
	TG (51:3)	144 ± 22	20 ± 7	156 ± 12	0.000	0.000	0.14	0.13
	TG (53:3)	133 ± 24	24 ± 8	134 ± 15	0.000	0.000	0.18	0.18
	TG (54:2)	443 ± 51	116 ± 34	410 ± 89	0.000	0.001	0.26	0.28
	TG (50:2)	1177 ± 140	356 ± 102	1290 ± 210	0.000	0.000	0.30	0.28
	TG (54:3)	1296 ± 143	430 ± 77	1288 ± 131	0.000	0.000	0.33	0.33
	TG (52:1)	231 ± 25	84 ± 23	203 ± 48	0.000	0.003	0.36	0.41
	TG (50:1)	562 ± 72	209 ± 63	579 ± 161	0.000	0.004	0.37	0.36
	TG (56:3)	163 ± 27	42 ± 22	148 ± 36	0.000	0.001	0.26	0.28
	TG (50:3)	830 ± 83	179 ± 67	945 ± 112	0.000	0.000	0.22	0.19
	DG (34:1)	184 ± 18	61 ± 18	191 ± 15	0.000	0.000	0.33	0.32
	TG (52:5)	489 ± 101	179 ± 50	549 ± 118	0.000	0.001	0.37	0.33
	TG (52:2)	3124 ± 323	1189 ± 347	3363 ± 272	0.000	0.000	0.38	0.35

Abbreviations: C = null control group, A = AMD-administered group, A+P = AMD plus probiotic blend-administered group. Metabolites are arranged in order of the magnitude of the A/C fold change.

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
