# Peer review of "Relationship Between the Gastrointestinal Side Effects of an Anti-Hypertensive Medication and Changes in the Serum Lipid Metabolome"

_nutrients, 2020, doi:10.3390/nu12010205_

Round 1

Reviewer 1 Report

[General comments]

This article has clarified that the unwanted side effects caused by ant-hypertensive drug is related with disorder of lipid metabolism, which may result in triggering inflammatory responses. The point is very important in warning the unlimited medication. In addition, it is interesting that the abnormalities induced by anti-hypertensive drug is mediated by the change of gut microbiome. The results are very clear and simple, but in order to confirm the benefits of probiotics, the authors had better check the reproducibility and the best condition for supplementation of probiotics.

[Specific comments]

The data of Figure 1 clearly indicated the separation between control/AMD+PB groups and AMD group. When seeing the number of symbols, it seems that there are 5 red triangles, 5 blue squares and 6 green circles. If this is correct, is there any reason that the data of one AMD rat and one AMD+PB rat were not included? In a previous report by authors (Benef Microbes 8:801-808, 2017), AMD was found to increase inflammatory cytokine (TNF-α, IL-1β and IL-6) levels in GI tract. And this article shows that AMD induces the change of lipid metabolites in serum 2 weeks after the treatment with AMD. Does the change of lipid metabolism precede the onset of inflammatory responses? Figures 3 and 4 clearly show that PB supplementation suppresses corticosterone production independent of ACTH axis. Do the authors consider the involvement of neurological control by PB or other mechanisms? Probiotic blend used in this experiment contains six bacterial strains and prebiotics such as fructooligosaccharide and galactooligosaccharide. To ameliorate the side effect of AMD, are all the components required, or is there any key component inevitable. Although the authors might have already confirmed that gut microbiome recovers by PB in AMD-treated rats in the previous report, it is better to re-check the association of gut microbiome recovery and normalization of lipid metabolism.

Author Response

We are grateful to the reviewers for the helpful comments on our manuscript, nutrients-667162 (Relationship between the Gastrointestinal Side Effects of an Anti-Hypertensive Medication and Changes in the Serum Lipid Metabolome). Our planned replies to the reviewers’ comments are described below.

The data of Figure 1 clearly indicated the separation between control/AMD+PB groups and AMD group. When seeing the number of symbols, it seems that there are 5 red triangles, 5 blue squares and 6 green circles. If this is correct, is there any reason that the data of one AMD rat and one AMD+PB rat were not included?

- In AMD group, one rat died before scarification.

In a previous report by authors (Benef Microbes 8:801-808, 2017), AMD was found to increase inflammatory cytokine (TNF-α, IL-1β and IL-6) levels in GI tract. And this article shows that AMD induces the change of lipid metabolites in serum 2 weeks after the treatment with AMD. Does the change of lipid metabolism precede the onset of inflammatory responses?

- Both blood and intestine tissue samples were taken after scarification for analysis.

Figures 3 and 4 clearly show that PB supplementation suppresses corticosterone production independent of ACTH axis. Do the authors consider the involvement of neurological control by PB or other mechanisms?

- Please understand that we refrained from considering the involvement of neurological control by PB or other mechanisms because it would be mostly speculation at this time.

Probiotic blend used in this experiment contains six bacterial strains and prebiotics such as fructooligosaccharide and galactooligosaccharide. To ameliorate the side effect of AMD, are all the components required, or is there any key component inevitable. Although the authors might have already confirmed that gut microbiome recovers by PB in AMD-treated rats in the previous report, it is better to re-check the association of gut microbiome recovery and normalization of lipid metabolism.

- We agree with the reviewer’s point of view that it is better to see whether all the components of the blend are required to ameliorate the side effect of AMD. As mentioned in our previous paper (Benef Microbes 8:801-808, 2017), we had previously conducted an experiment to test the same blend for its efficacy in an obesity model by using the same excipients of the blend as a negative control and found that no significant effect on the gut microbiota was induced by the excipients although we have not checked for the respective effect of each strain. Please understand that the main point of our present work is not to explore detailed mechanism, but to provide a clue for better understanding of the roles of gut bacteria in host metabolism.

Reviewer 2 Report

Line 42. “100–400 trillion” it is better to use 1-4 x 10^15. Line 44. GItract. Please separate the words Figure 5. The quality should be improved. The text should be increased in order to be easily readable. Figure 3. Please correct. AMD+PB. Please correct it in all manuscript since in several parts there are AMD+P Conclusions should be increased and include more results of the present study. The authors should add discussion regarding the effect of PB. Almost nothing is written or discussed. Otherwise delete this experiments and results. What did the authors expect by the addition of PB? In general more discussion is needed by comparing the results of the present study with similar other studies.

Author Response

We are grateful to the reviewers for the helpful comments on our manuscript, nutrients-667162 (Relationship between the Gastrointestinal Side Effects of an Anti-Hypertensive Medication and Changes in the Serum Lipid Metabolome). Our planned replies to the reviewers’ comments are described below.

Line 42. “100–400 trillion” it is better to use 1-4 x 10^15.

- We have changed it according to your suggestion.

Line 44. GItract.

- We have corrected it according to your suggestion.

Please separate the words Figure 5. The quality should be improved. The text should be increased in order to be easily readable.

- We have changed it according to your suggestion.

Figure 3. Please correct. AMD+PB. Please correct it in all manuscript since in several parts there are AMD+P

- We have changed it according to your suggestion.

Conclusions should be increased and include more results of the present study.

- We have revised the manuscript according to your suggestion.

The authors should add discussion regarding the effect of PB. Almost nothing is written or discussed. Otherwise delete this experiments and results. What did the authors expect by the addition of PB? In general more discussion is needed by comparing the results of the present study with similar other studies.

- We have revised the manuscript according to your suggestion.

Round 2

Reviewer 2 Report

Line 43 It is not correct. Trillion is not 105

In my previous email: "The authors should add discussion regarding the effect of PB. Almost nothing is written or discussed. Otherwise delete this experiments and results. What did the authors expect by the addition of PB? In general more discussion is needed by comparing the results of the present study with similar other studies."

There is nothing new about my comment in the revised version.

Author Response

I wish to express my gratitude to you for the helpful comments on our manuscript, nutrients-667162 (Relationship between the Gastrointestinal Side Effects of an Anti-Hypertensive Medication and Changes in the Serum Lipid Metabolome). My response to your comments is as follows.

Line 43 It is not correct. Trillion is not 105

- I have corrected the typo.

The authors should add discussion regarding the effect of PB. Almost nothing is written or discussed. Otherwise delete this experiments and results. What did the authors expect by the addition of PB? In general more discussion is needed by comparing the results of the present study with similar other studies.

- I have revised the manuscript according to your suggestion (lines 198-204).